# Prevalence and access to care for cardiovascular risk factors in older people in Sierra Leone: a cross-sectional survey

Maria Lisa Odland [ID] ,[1] Tahir Bockarie,[2] Haja Wurie,[3] Rashid Ansumana,[4,5] Joseph Lamin,[4] Rachel Nugent,[6] Ioannis Bakolis,[7,8] Miles Witham,[9,10] Justine Davies [ID] [1,11]

For numbered affiliations see end of article.

**Correspondence to**
Dr Maria Lisa Odland;
m.l.odland@bham.ac.uk

## ABSTRACT

**Introduction** Prevalence of cardiovascular disease risk factors (CVDRFs) is increasing, especially in low-income countries. In Sierra Leone, there is limited empirical data on the prevalence of CVDRFs, and there are no previous studies on the access to care for these conditions.

**Methods** This study in rural and urban Sierra Leone collected demographic, anthropometric measurements and clinical data from randomly sampled individuals over 40 years old using a household survey. We describe the prevalence of the following risk factors: diabetes, hypertension, dyslipidaemia, overweight or obesity, smoking and having at least one of these risk factors. Cascades of care were constructed for diabetes and hypertension using % of the population with the disease who had previously been tested ('screened'), knew of their condition ('diagnosed'), were on treatment ('treated') or were controlled to target ('controlled'). Multivariable regression was used to test associations between prevalence of CVDRFs and progress through the cascade for hypertension with demographic and socioeconomic variables. In those with recognised disease who did not seek care, reasons for not accessing care were recorded.

**Results** Of 2071 people, 49.6% (95% CI 49.3% to 50.0%) of the population had hypertension, 3.5% (3.4% to 3.6%) had diabetes, 6.7% (6.5% to 7.0%) had dyslipidaemia, 25.6% (25.4% to 25.9%) smoked and 26.5% (26.3% to 26.8%) were overweight/obese; a total of 77.1% (76.6% to 77.5%) had at least one CVDRF. People in urban areas were more likely to have diabetes and be overweight than those living in rural areas. Moreover, being female, more educated or wealthier increased the risk of having all CVDRFs except for smoking. There is a substantial loss of patients at each step of the care cascade for both diabetes and hypertension, with less than 10% of the total population with the conditions being screened, diagnosed, treated and controlled. The most common reasons for not seeking care were lack of knowledge and cost.

**Conclusions** In Sierra Leone, CVDRFs are prevalent and access to care is low. Health system strengthening with a focus on increased access to quality care for CVDRFs is urgently needed.

## Strengths and limitations of this study

► This study was adequately powered to detect cardiovascular risk factors in this population.
► We used random sampling and probability weights to avoid potential biases.
► The data collection was limited to one district in Sierra Leone.
► We did not control for clustering at household level as few houses supplied more than one participant.
► Clinical diagnoses in this study were defined for the purpose of this study based on measurements taken at a single point in time.

## INTRODUCTION

Non-communicable diseases (NCDs) such as cardiovascular disease and its risk factors are major health problems globally.[1] The reduction in deaths from infections including HIV has led to an ageing population, which has, together with lifestyle transitions towards a high-calorie, low-activity and urban lifestyle, led to a high and rising prevalence of NCDs in lower and middle-income countries (LMICs).[2–4] In fact, high blood pressure has become the largest contributor to premature mortality globally,[3 4] and cardiovascular diseases (including coronary heart disease and stroke) are the most common NCDs, globally responsible for an estimated 17.8 million deaths in 2017.[2] More than three quarters of these were in LMICs.[2]

However, surveillance of the prevalence of cardiovascular disease risk factors (CVDRFs) is very limited in the poorest countries in the world. Sierra Leone is a low-income country situated in West Africa. It has a human development index of 0.419 (184 of 189 countries) and a maternal mortality ratio (1360 per 100 000 live births) and under-5 mortality rate (110.5 per 1000 live births) among the

highest in the world.[5] The civil war from 1991 to 2002 disrupted infrastructure development, including that of the health system. Moreover, the 2013–2016 Ebola virus disease created a public health crisis and drew resources away from broader development of the health system.[6 7]

In recent years, both gross domestic product and life expectancy at birth have increased in Sierra Leone.[8] In other countries (including those in sub-Saharan Africa) that have undergone a demographic transition, it has been accompanied by an increasing burden of CVDRFs—such as diabetes and hypertension, dyslipidaemia and overweight—with consequent macrovascular and microvascular disease outcomes, such as heart attacks, strokes and blindness.[2 7] Unfortunately, although estimates of CVDRF prevalence from modelling studies exist, very little systematic, direct measurement of the burden of CVDRFs in the country has occurred; although small outdated studies have suggested a high burden of CVDRFs.[9–13] These other studies are either more than 10 years old or have fewer than 700 participants. Additionally, there is no information on whether and how sufferers are accessing care. Sierra Leone is developing its national policy and strategic plan for NCDs. To ensure efficient use of the already stretched healthcare resources, the strategic plan and its implementation needs to be informed by empirical information on the burden of risk factors and current access to care.[14] In order to provide evidence to assist health policy planning, this study aimed to describe the prevalence of CVDRFs in people over 40 years old in Sierra Leone, access to care for those risk factors and sociodemographic characteristics associated with CVDRFs and access to care.

## METHODS
### Study setting
The study was conducted in the district of Bo, located in the Southern Province of Sierra Leone, and one of 16 districts in the country. It has well-documented rural and urban areas and contains Sierra Leone's second largest city, Bo (online supplementary appendix figure 1).[15] The demographics, socioeconomic circumstances and geographical distribution of the population are similar to the larger Sierra-Leonean population.[15] In the last census in 2015, there were 575 478 inhabitants of Bo district, with 66.1% (380 307) living in rural areas and 33.9% living in urban areas, mostly in Bo City. Further, 17.4% (100 188) of the population are over 40 years of age.[15] Bo District has a mainly agriculture-based economy, but service-based industries are growing. Mende is the most used language, but Krio and English are also spoken.

### Sampling strategy
A sample size of 1893 participants was targeted to allow detection of diabetes prevalence (the risk factor thought likely to have the lowest prevalence) of 4% with a precision of ±1%. To allow for non-response and non-availability of data, we oversampled by 20%. A sampling of individuals over 40 years of age was done from rural or urban areas

in proportion with known patterns from the 2015 populations and housing census of habitation of these areas in the over 40s.[15] The 15 rural chiefdoms that comprise Bo District were listed in alphabetical order and 7 chiefdoms with separate geographical locations were chosen for the study using random number generator. Settlement groups or villages within these chiefdoms were identified and two were randomly chosen for study. Seven urban communities were randomly selected from 24 urban communities using similar methods of selection. Numbers of participants to sample from urban and rural areas were calculated based on the proportions of people living in these areas. In each urban community, numbers needed to study was 100. In each rural settlement or village, numbers needed to study was 93. If numbers were not achieved in the two selected areas, the next randomly ordered one was selected for study. Census information was not detailed enough to allow further identification of households with residents over 40 years old. Thus, data collection proceeded in each urban subdistrict or village, with data collectors starting at random points within each area and walking along a road or track sampling from every second household. Each household was permitted to enter no more than two people over 40 into the study. In villages where there were 93 households or fewer, all households were sampled. The geographical radius of the study was limited to 40 km from the centre of Bo to ensure accessibility. All chiefdoms and subdistricts in Bo were represented within this radius.

### Data collection
Data were collected electronically by trained staff using the ODK (Open Data Kit) platform[16] from September to November 2018. The survey questionnaire was written in English but interviews were conducted in one of the local languages, either Krio or Mende.

Survey questions asked about sociodemographic information: gender, age, highest level of education completed (no formal schooling, primary, junior secondary, senior secondary, higher education or refused), employment in the past 12 months (as government employee, non-government employee, self-employed, non-paid worker, student, homemaker, retired, unemployed able to work, unemployed unable to work or refused) and marital status (as single, cohabiting, currently married, multiple partners, divorced, widowed or refused). There were also 49 questions on household assets and construction materials. Questions on smoking, awareness of the presence of CVDRF and whether respondents were on treatment for these risk factors were based on the WHO STEPS survey; for those who reported suffering from a CVDRF, or had had a stroke, heart attack or angina, whether care had been accessed, where care was accessed and reasons for not accessing care were also asked.[11]

Height was measured using a tape with participants standing with their backs, hips and heels against a wall and looking ahead horizontally (this method was validated using a Height Measure (SECA 213) during training).

An Accuweight digital body scale was used for measuring weight while wearing light clothing and without shoes.

Sitting blood pressure was measured using an Omron M6 AC LED blood pressure monitor. Three measurements were taken with 5 min intervals between measurements. Blood samples were taken first thing in the morning after an 8-hour overnight fast. Glucose and cholesterol were measured using the Accutrend Plus Blood Test Metre (Diagnostics Roche) point-of-care device.

Participant's fasting status was checked prior to the blood sample being taken, and those who reported not fasting were labelled as such. Cholesterol samples were obtained from every second participant, while glucose was measured from all participants. The conversion rate of 1.11 was used to convert capillary glucose to plasma glucose.[17]

### Outcome measures

Body mass index (BMI) was defined as weight (measured in kilograms (kg)) divided by height (measured in metres squared) and classified as normal weight ($<25$ kg/m$^2$) or overweight/obese (BMI $\geq 25$ kg/m$^2$). An additional analysis with normal and overweight ($<30$ kg/m$^2$) versus obese (BMI $\geq 30$ kg/m$^2$) was also done. Diabetes was defined as fasting plasma glucose $\geq 7.0$ mmol/L (126 mg/dL) or as random plasma glucose $\geq 11.1$ mmol/L (200 mg/dL). Hypertension was defined as recorded systolic blood pressure $\geq 140$ or diastolic $\geq 90$ mm Hg, calculated using the average of the final two readings. Dyslipidaemia was defined as measured total cholesterol level $\geq 6.21$ mmol/L, or low-density lipoprotein $\geq 4.1$ mmol/L, or high-density lipoprotein $<1.19$ mmol/L. Participants that reported they had taken drugs for diabetes, hypertension or dyslipidaemia within the last 2 weeks were classified as having these conditions irrespective of their biomarker measurements. Smoking was defined as current smoker if participants either reported currently smoking or had ceased within the last year, or non-smoking for others. Educational level was defined as having completed 'any level of education' (primary, secondary or university) or 'no completed education'. Marital status was defined as married/cohabiting or single/widowed/divorced. Wealth quintiles were derived from the first principal component of household assets and construction materials using the method of Filmer and Pritchett.[18]

### Access to healthcare
#### Self-reported access to care

Everyone with self-reported previous diagnosis of hypertension, diabetes, dyslipidaemia, angina, heart attack or stroke was asked if they had accessed care for their conditions in the last 4 weeks or 3 months. Reasons for not accessing care were explored for the ones who did not have self-reported access to care.

#### Construction of the care cascade

A cascade of care was constructed for diabetes and hypertension. The stages in the care cascade are as follows:

1. Prevalent disease (the population defined as having hypertension or diabetes).
2. Ever been screened (the population who have had their blood pressure or glucose measured by a health personnel).
3. Prior diagnosis (the population who have ever been told by a doctor or other healthcare worker that they have hypertension or diabetes).
4. Currently on treatment (the population who have taken drugs for hypertension or diabetes in the last 2 weeks).
5. Disease control (the population who have their condition controlled to target at study measurement).

Entry into each subsequent stage of the cascade was contingent on an individual having achieved the previous stage. The population prevalence for diabetes and hypertension formed the denominators for all other stages of the respective care cascade. Additionally, the loss from each step in the care cascade was calculated using the people who had achieved the previous step as the denominator.

### Statistical analysis

Statistical analysis was done using SPSS V.24 (IBM). Descriptive statistics were described using mean and SD for normally distributed continuous variables and median and IQR for non-normally distributed variables. Univariate associations between independent variables (demographic characteristics) and outcomes (CVDRFs) were tested using $\chi^2$ tests and Kendalls Tau-B for categorical variables and Mann-Whitney and Spearman's Rho for continuous variables. Multivariable analyses were performed using binary logistic regression with forced entry of all independent variables. For hypertension, factors associated with achieving each step in the cascade were tested. This was not done for diabetes as numbers were too small for meaningful results. A sensitivity analysis using BMI>30 as a cut-off was done (online supplementary appendix table 1), and we decided to use age as categorical variable in the multivariable analysis due to non-linear association with some outcomes (eg, demographic characteristics and CVDRF). Confidence intervals (CIs) for proportions were calculated according to a method described by Robert Newcombe derived from a procedure outlined by Wilson.[19]

Probability weights for age and sex in Bo-South were calculated based on the 2015 Population and Household Census.[15] All analyses were done using weight adjustments. Clustering at village level was adjusted for in the multivariable analyses.

### Patient and public involvement statement

Participants were not directly involved in planning the study.

### RESULTS

The final sample included 2071 individuals. The weighted demographic characteristics and prevalence of cardiovascular risk factors of the study population are presented in table 1. The unweighted proportions of demographic characteristics of participants with measured cholesterol versus not measured cholesterol are presented in online supplementary

**Table 1** Weighted demographic characteristics and prevalence of cardiovascular risk factors in Bo, Sierra Leone (n=2071)

| Parameter | Group | % using weights |
|---|---|---|
| Place of living | Rural | 62.9 |
| | Urban | 37.1 |
| Gender | Female | 49.0 |
| | Male | 51.0 |
| Age median (IQR), n=2062 | Years | 51.0 (45.0–63.0) |
| Education level, n=2070 | No completed education | 67.4 |
| | Any education | 32.6 |
| Marital status, n=2069 | Married/cohabiting | 72.6 |
| | Single/widowed/divorced | 27.4 |
| Wealth quintile, n=1991 | 1 | 20.5 |
| | 2 | 20.5 |
| | 3 | 20 |
| | 4 | 19.9 |
| | 5 | 19.1 |
| Cardiovascular disease risck factors (CVDRFs) | Hypertension, n=2070 | 49.6 |
| | Mean (SD) Systolic blood pressure | 136.19 (25.24) |
| | Mean (SD) Diastolic blood pressure | 87.52 (14.11) |
| | Diabetes, n=2019 | 3.5 |
| | Dyslipidaemia, n=840 | 6.7 |
| | Overweight/obesity, n=1947 | 26.5 |
| | Smoking | 25.6 |
| | One CVD risk factor or more out of a possible 7, including cholesterol (n = 789) | Including cholesterol 77.1 |
| | One CVD risk factor or more out of a possible 6, excluding cholesterol (n = 1896) | Excluding cholesterol 74.5 |

appendix table 2. Those who had their cholesterol measured were similar to those who did not. However, there were fewer males who had cholesterol measured.

### Population characteristics and risk factor prevalence

The population predominately lived in rural areas (62.9%) and 49.0% of the study population was female. The median age was 51.0 years, 67.4% had not completed any education and 72.6% were married/cohabiting. The prevalence of hypertension was 49.6% (95% CI 49.3% to 50.0%), while the prevalence of diabetes and dyslipidaemia were 3.5% (95% CI 3.4% to 3.6%) and 6.7% (95% CI 6.5% to 7.0%), respectively. Overweight or obesity (BMI $\geq$25 kg/m$^2$) was present in 26.5% (95% CI 26.3% to 26.8%) of the study population and 25.6% (95% CI 25.4% to 25.9%) of the participants were current or recent (within the last year) smokers. Altogether, 77.1% (95% CI 76.6% to 77.5%) of the study population had at least one CVDRF when including cholesterol (and limiting the denominator to those 789 who had cholesterol measured), while when excluding cholesterol as a variable (and with a denominator of 1896 who had information on all other CVDRFs) the prevalence of at least one CVDRF was 74.5% (95 CI 74.3% to 74.8%). Univariate associations

between demographic characteristics and CVDRF are presented in online supplementary appendix table 3.

In the multivariable analysis (table 2), living in an urban area was independently associated with all CVDRFs except for dyslipidaemia (which was more prevalent in those living in rural areas). Male sex was independently associated with lower prevalence of CVDRFs with the exceptions of smoking and the presence of any risk factor. Increasing age was independently associated with increasing prevalence of hypertension, diabetes or dyslipidaemia and with a decreased prevalence of being overweight or smoking. The prevalence of CVDRFs according to age group and sex is shown in figure 1. Having any education compared with no complete education was independently associated with increased prevalence of all CVDRFs expect for smoking. Being married or cohabiting was independently associated with lower prevalence of all CVDRFs except for diabetes and obesity. Wealth remained independently associated with all CVDRFs except for smoking, where increasing wealth quintile was associated with a lower prevalence of smoking.

### Access to healthcare

A total of 496 participants reported a previous diagnosis of hypertension, diabetes or dyslipidaemia, angina, heart

**Table 2** Multivariable associations between demographic characteristics and cardiovascular risk factors including cholesterol (n=2071)

| Parameter | Group | Hypertension OR (95% CI) | P value | Diabetes OR (95% CI) | P value | Dyslipidaemia OR (95% CI) | P value | Overweight/obese OR (95% CI) | P value | Smoking OR (95% CI) | P value | Total CVD risk factors incl. chol OR (95% CI) | P value | Total CVD risk factors exl. chol OR (95% CI) | P value |
|---|---|---|---|---|---|---|---|---|---|---|---|---|---|---|---|
| Place of living | Rural | Referent | – | Referent | – | Referent | – | Referent | – | Referent | – | Referent | – | Referent | – |
| | Urban | 1.04 (1.01 to 1.08) | 0.014 | 1.46 (1.34 to 1.60) | <0.001 | 0.84 (0.75 to 0.93) | 0.001 | 1.17 (1.12 to 1.21) | <0.001 | 1.13 (1.08 to 1.17) | <0.001 | 0.99 (0.93 to 1.05) | 0.614 | 1.06 (1.02 to 1.10) | 0.002 |
| Gender | Female | Referent | – | Referent | – | Referent | – | Referent | – | Referent | – | Referent | – | Referent | – |
| | Male | 0.78 (0.75 to 0.80) | <0.001 | 0.75 (0.69 to 0.82) | <0.001 | 0.88 (0.80 to 0.97) | 0.013 | 0.31 (0.30 to 0.32) | <0.001 | 9.15 (8.76 to 9.54) | <0.001 | 1.60 (1.52 to 1.70) | <0.001 | 1.43 (1.38 to 1.48) | <0.001 |
| Age | 40–49 | Referent | – | Referent | – | Referent | – | Referent | – | Referent | – | Referent | – | Referent | – |
| | 50–59 | 1.75 (1.69 to 1.81) | <0.001 | 2.10 (1.91 to 2.32) | <0.001 | 1.38 (1.25 to 1.53) | <0.001 | 0.84 (0.81 to 0.87) | <0.001 | 0.84 (0.81 to 88.0) | <0.001 | 0.93 (0.88 to 0.99) | 0.023 | 1.15 (1.11 to 1.20) | <0.001 |
| | 60–69 | 2.35 (2.26 to 2.45) | <0.001 | 2.77 (2.50 to 3.07) | <0.001 | 1.36 (1.22 to 1.53) | <0.001 | 0.85 (0.81 to 0.89) | <0.001 | 0.58 (0.56 to 0.61) | <0.001 | 1.25 (1.16 to 1.35) | <0.001 | 1.70 (1.60 to 1.81) | <0.001 |
| | 70–79 | 3.43 (3.27 to 3.61) | <0.001 | 3.46 (3.07 to 3.89) | <0.001 | 1.76 (1.52 to 2.05) | <0.001 | 0.70 (0.66 to 0.75) | <0.001 | 0.36 (0.33 to 0.38) | <0.001 | 2.24 (2.00 to 2.51) | <0.001 | 1.25 (1.16 to 1.34) | <0.001 |
| | >80 | 3.13 (2.96 to 3.32) | <0.001 | 1.76 (1.69 to 1.99) | <0.001 | 0.98 (0.81 to 1.19) | 0.835 | 0.49 (0.45 to 0.53) | <0.001 | 0.52 (0.48 to 0.56) | <0.001 | 1.22 (1.09 to 1.38) | 0.001 | 1.07 (1.16 to 1.34) | <0.001 |
| Education level | No complete education | Referent | – | Referent | – | Referent | – | Referent | – | Referent | – | Referent | – | Referent | – |
| | Any education | 1.17 (1.14 to 1.21) | <0.001 | 1.83 (1.69 to 1.99) | <0.001 | 1.08 (0.98 to 1.18) | 0.111 | 1.63 (1.57 to 1.69) | <0.001 | 0.86 (0.83 to 0.89) | <0.001 | 0.91 (0.86 to 0.96) | 0.001 | 1.07 (1.04 to 1.11) | <0.001 |
| Marital status | Single/divorced/widow | Referent | – | Referent | – | Referent | – | Referent | – | Referent | – | Referent | – | Referent | – |
| | Married/Cohabiting | 0.80 (0.78 to 0.83) | <0.001 | 1.01 (0.93 to 1.11) | 0.785 | 0.62 (0.56 to 0.68) | <0.001 | 1.10 (1.06 to 1.15) | <0.001 | 0.80 (0.77 to 0.84) | <0.001 | 0.81 (0.76 to 0.86) | <0.001 | 0.84 (0.81 to 0.88) | <0.001 |
| Wealth quintile | 1 | Referent | – | Referent | – | Referent | – | Referent | – | Referent | – | Referent | – | Referent | – |
| | 2 | 0.83 (0.79 to 0.86) | <0.001 | 1.31 (1.14 to 1.52) | <0.001 | 1.84 (1.47 to 2.29) | <0.001 | 1.42 (1.34 to 1.50) | <0.001 | 0.96 (0.92 to 1.01) | 0.105 | 1.06 (0.98 to 1.14) | 0.142 | 1.10 (1.05 to 1.15) | <0.001 |
| | 3 | 0.99 (0.95 to 1.03) | 0.698 | 1.20 (1.04 to 1.39) | 0.014 | 2.36 (1.90 to 2.29) | <0.001 | 1.66 (1.57 to 1.76) | <0.001 | 0.71 (0.68 to 0.75) | <0.001 | 0.87 (0.81 to 0.94) | 0.001 | 0.91 (0.87 to 0.95) | 0.001 |
| | 4 | 1.30 (1.25 to 1.36) | <0.001 | 1.64 (1.42 to 1.88) | <0.001 | 6.19 (5.07 to 7.56) | <0.001 | 3.00 (2.84 to 3.16) | <0.001 | 0.51 (0.49 to 0.54) | <0.001 | 1.41 (1.30 to 1.53) | <0.001 | 1.32 (1.25 to 1.38) | <0.001 |
| | 5 | 1.60 (1.52 to 1.69) | <0.001 | 2.70 (2.34 to 3.12) | <0.001 | 11.16 (9.05 to 13.76) | <0.001 | 5.11 (4.81 to 5.44) | <0.001 | 0.39 (0.37 to 0.42) | <0.001 | 2.46 (2.23 to 2.73) | <0.001 | 1.62 (1.53 to 1.72) | <0.001 |

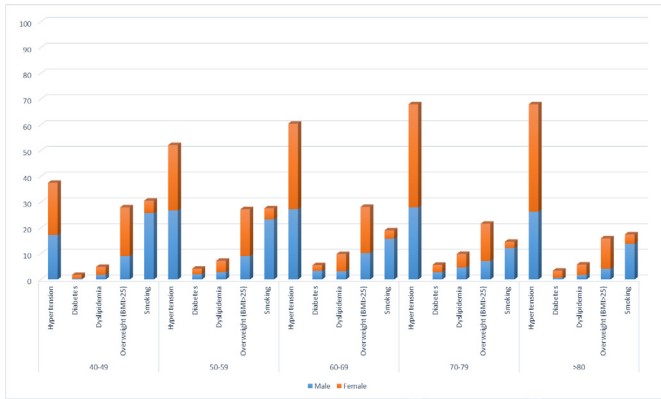

**Figure 1** Prevalence of cardiovascular risk factors according to age and sex. BMI, body mass index.

attack or stroke. Of these, only 88 (17.7%) stated that they had accessed healthcare for their cardiovascular diseases in the last 3 months and only 8.9% had accessed healthcare in the last 4 weeks. The most common reasons for not accessing healthcare were thinking that it was not necessary (47.0%) or that it was too expensive (24.5%). Everyone who accessed care in the last 3 months visited a modern health facility, with 35.5% visiting community-based health service and 63.2% a hospital-based health service. Nobody reported having visited a traditional healer for their condition.

The cascade of care for hypertension is shown in figure 2. Among those with hypertension, 59.2% reported that they had their blood pressure measured by a healthcare professional (screened), and 33.2% had ever been diagnosed with hypertension. There was a substantial loss to care at both steps, 40.8% and 44.0%, respectively. Only 14.7% of people with hypertension were currently on treatment (taken medication for hypertension in the last 2 weeks), and of the people who were currently on treatment, 31.2% achieved control. The last step of the cascade, being controlled, had the biggest loss to care from the previous step of 68.8%. In the multivariable analysis of the hypertension cascade (table 3), people

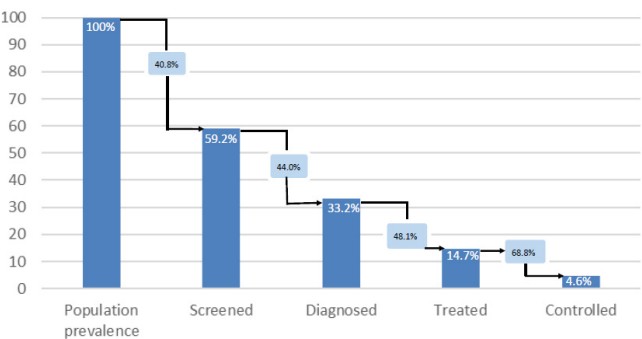

**Figure 2** Cascade of care for hypertension using % of the population with the disease who had previously been tested ('screened'), knew of their condition ('diagnosed'), were on treatment ('treated') or were controlled to target ('controlled'). The loss to care at each step is described by the black arrows.

living in an urban area were significantly more likely to pass through all the steps of the cascade apart from being diagnosed. Women were more likely than men to be screened or diagnosed, but not treated; men were more likely than women to be controlled. There was no clear relationship between age groups and progress through the cascade. Having some education or being wealthier were significantly associated with passing through the first three steps of the cascade, but not with being controlled.

The cascade of care for diabetes is presented in figure 3. Out of all the people with diabetes in our study population (hyperglycaemic on measurement or taken medication in the last 2 weeks), the largest loss to care was at the stage of screening, with only 43.0%% of participants reporting that they had had their blood sugar measured at any time previously. There was a more modest loss to care for the next step with 32.9% of the participants with diabetes reporting that they had ever been told that they have diabetes. For the next step, only 19.0% of the participants with diabetes reported that they had been taking treatment for diabetes in the last 2 weeks. Finally, 8.6% of the total population with diabetes had achieved control of their disease which is less than half the population that reported that they were on treatment. For diabetes, the sample size was too small to do multivariable analysis with demographic characteristics in the different steps in the cascade.

## DISCUSSION

This paper reports one of the first studies to provide estimates of the prevalence of all CVDRFs in Sierra Leone; it is the first that we are aware of to publish on access to care for CVDRFs. Our data suggest that the prevalence of CVDRFs in Sierra Leone is high, with 75% of the population over 40 having at least one CVDRF. The risk of having a CVDRF increased with age, and CVDRF was more common in the urban population, among women, unmarried people and individuals with education and in the highest wealth quintile. Smoking was very common among men, giving them a higher overall risk of having at least one CVDRF. Also, our analysis revealed that there are very high rates of unmet need for hypertension and diabetes care. Less than 20% of the population with hypertension, diabetes and dyslipidaemia accessed healthcare in the last 3 months.

Although we sampled only one area in Sierra Leone, the population structure is similar to other areas in Sierra Leone except for Freetown.[15] Thus, our findings give insight into the likely prevalence and associations across the country. Indeed, our estimate of hypertension of about 50% is similar to that found previously in Sierra Leone in the same age group in other areas.[9 10] There are very little data available on diabetes from Sierra Leone, but the most recent estimates, both empirical and modelled, were much higher than we found in our study.[12 20] For example, the NCD Risk collaboration estimated prevalence of diabetes to be 7.1% (95% CI 3.55 to 12.15) in

**Table 3** Multivariate associations between demographic characteristics and access to care for hypertension for people with hypertension (n=1092)

| Parameter | Group | Screened (n=1092) OR (95% CI) | P value | Diagnosis (n=646) OR (95% CI) | P value | Treated (n=362) OR (95% CI) | P value | Controlled (n=160) OR (95% CI) | P value |
|---|---|---|---|---|---|---|---|---|---|
| Place of living | Rural | Referent | – | Referent | – | Referent | – | Referent | – |
| | Urban | 1.61 (1.53 to 1.68) | <0.001 | 0.97 (0.91 to 1.03) | 0.325 | 1.36 (1.23 to 1.50) | <0.001 | 2.13 (1.77 to 2.58) | <0.001 |
| Gender | Female | Referent | – | Referent | – | Referent | – | Referent | – |
| | Male | 0.70 (0.67 to 0.73) | <0.001 | 0.79 (0.74 to 0.84) | <0.001 | 1.13 (1.02 to 1.25) | 0.015 | 1.12 (0.97 to 1.30) | 0.121 |
| Age | 40–49 | Referent | – | Referent | – | Referent | – | Referent | – |
| | 50–59 | 0.91 (0.87 to 0.96) | 0.001 | 1.36 (1.28 to 1.45) | <0.001 | 1.64 (1.49 to 1.80) | <0.001 | 0.95 (0.82 to 1.11) | 0.535 |
| | 60–69 | 1.49 (1.41 to 1.58) | <0.001 | 1.01 (0.94 to 1.07) | 0.874 | 2.15 (1.93 to 2.39) | <0.001 | 0.89 (0.76 to 1.04) | 0.141 |
| | 70–79 | 1.22 (1.14 to 1.30) | <0.001 | 0.55 (0.51 to 0.60) | <0.001 | 1.44 (1.25 to 1.65) | <0.001 | 1.13 (0.93 to 1.38) | 0.221 |
| | >80 | 0.63 (0.59 to 0.68) | <0.001 | 0.72 (0.64 to 0.80) | <0.001 | 1.55 (1.29 to 1.87) | <0.001 | 1.09 (0.84 to 1.42) | 0.516 |
| Education level | No complete education | Referent | – | Referent | – | Referent | – | Referent | – |
| | Any education | 1.78 (1.69 to 1.86) | <0.001 | 1.09 (1.03 to 1.16) | 0.002 | 2.93 (2.69 to 3.18) | <0.001 | 0.70 (0.62 to 0.80) | <0.001 |
| Marital status | Single/divorced/widow | Referent | – | Referent | – | Referent | – | Referent | – |
| | Married/cohabiting | 1.02 (0.97 to 1.07) | 0.465 | 0.95 (0.89 to 1.01) | 0.12 | 0.67 (0.61 to 0.73) | <0.001 | 1.17 (1.02 to 1.33) | 0.022 |
| Wealth quintile | 1 | Referent | – | Referent | – | Referent | – | Referent | – |
| | 2 | 1.22 (1.15 to 1.30) | <0.001 | 1.59 (1.45 to 1.75) | <0.001 | 2.40 (1.97 to 2.91) | <0.001 | 0.25 (0.18 to 0.37) | <0.001 |
| | 3 | 1.77 (1.67 to 1.88) | <0.001 | 1.63 (1.50 to 1.79) | <0.001 | 2.95 (2.47 to 3.53) | <0.001 | 0.17 (0.12 to 0.22) | <0.001 |
| | 4 | 2.63 (2.47 to 2.80) | <0.001 | 1.45 (1.33 to 1.58) | <0.001 | 5.37 (4.52 to 6.39) | <0.001 | 0.16 (0.12 to 0.22) | <0.001 |
| | 5 | 4.21 (3.91 to 4.54) | <0.001 | 2.24 (2.04 to 2.46) | <0.001 | 4.97 (4.16 to 5.93) | <0.001 | 0.28 (0.20 to 0.39) | <0.001 |

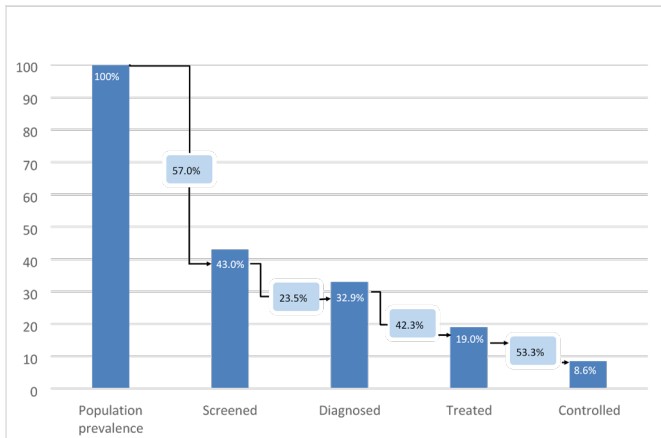

**Figure 3** Cascade of care for diabetes using % of the population with the disease who had previously been tested ('screened'), knew of their condition ('diagnosed'), were on treatment ('treated') or were controlled to target ('controlled'). The loss to care at each step is described by the black arrows.

2014.[20] The prevalence of diabetes in urban areas in our material (5.5%) was, however, similar to a previous study (6.2%) from 2012 to 2014 collected in only urban areas of Bo.[12] An older study conducted in Bo in 1997 reported a lower prevalence of 2.4% in the urban population and 0% in the rural population.[21] Diabetes prevalence might be rising with time, but the methodologies used in the previous studies make comparisons difficult. Both the previous studies were also much smaller in sample size (n=694 and n=501) than ours, and likely underpowered.

In contrast, the prevalence of hypertension in our study is higher than previous empirical data from the WHO STEPS survey conducted in 2009, and which found hypertension in 37% of males and 33% of females.[11] The population sampled in the previous WHO STEPS survey was younger (25–65 years) than in our study though, and the prevalence of hypertension is also likely to have increased in the past years.

Other areas in West Africa have also reported a similarly high prevalence of CVDRFs to what we have found, although prevalence of hypertension in Sierra Leone in our study is higher than other regional estimates from countries like Nigeria and Ghana.[3 22–25] Twenty-five per cent of the population in our sample were overweight or obese, which is surprising for one of the poorest countries in the world. However, our estimates of overweight/obesity are slightly lower than those derived from the WHO STEPS survey from 2009,[11] and lower than those reported from Nigeria, so it is unlikely that our findings overestimate the prevalence.[24] The geographical and socioeconomic and education balance of most CVDRF that we found are also reflective of findings from other studies in the region.[24 25] However, in other studies, CVDRFs like diabetes and hypertension are more prevalent in males in contrast to our findings.[24 25] Still, overall, males actually have a higher risk of having at least one

CVDRF than females in our sample. This makes men a vulnerable group when it comes to CVDRFs, especially since the cascade analysis suggests that they are less likely to enter into the healthcare system for their conditions than women.

The low prevalence of people with hypertension being controlled for their condition is similar to what has been previously shown in countries in sub-Saharan Africa.[4] Regarding diabetes, other studies have shown that many low-income countries in sub-Saharan Africa perform better than Sierra Leone on access to care with an average of more than 15%–20% of the patients achieving control of the disease.[26 27] However, similar to our findings, the biggest loss to care was at the stage of screening.[26] Although there are no studies done on the access to care for CVDRFs in Sierra Leone, previous studies on HIV care have shown that the loss to care is substantial with only 22.8% of patients with newly diagnosed HIV receiving effective treatment.[28] It might be tenuous to compare HIV care and care for CVDRF, as HIV care receives substantial financial support from donors. Care for HIV is also largely separated from the public healthcare system, and health-seeking behaviour for HIV is affected by stigma. Nevertheless, it is another indication that the health system in Sierra Leone finds it challenging to provide long-term follow-up care for patients with chronic disorders.

Living in an urban area was a strong predictive factor for passing through the cascade steps and achieving control of hypertension. Women were more likely to be screened and diagnosed for hypertension than men which could be due to women accessing maternal and child healthcare (which has been a focus of healthcare efforts in Sierra Leone), gender norms and facility opening hours. It is important to ensure that efforts are made to encourage and retain men in care. People with higher education and in the highest wealth quintile were also more likely to access care; similar to previous findings regarding access to hypertension care in LMICs.[4 29] Poorer and uneducated people are also more likely to experience catastrophic health expenditure on accessing care for NCDs, and investments in improving hypertension care present an opportunity to reduce health inequalities between socioeconomic groups. Even if healthcare is free, which in Sierra Leone is the case for the 'destitute', Ebola survivors, pregnant women, lactating women or children under 5,[30] accessing care still requires transport costs and is time lost from income-generating activity.[31] That we found that the most common reasons for not accessing care included cost suggests that addressing this barrier is key to providing care for sufferers of CVDRF in Sierra Leone. Interestingly, the people most likely to access care in our study (high education and wealth) were less likely to succeed at the last step in the cascade by achieving control of their condition. One reason for this could be that medications are not taken regularly. However, this finding could also be due to lack of study power due to the low number of people reaching the last step in the cascade.

This study is one of the first studies to report prevalence of multiple CVDRFs in such a large sample from Sierra Leone and the first study to report access to care for these. The study sample is larger than any previous studies on CVDRF in Sierra Leone, and the data sampling and analysis were done in a rigorous way to avoid potential biases. Bo also consists of urban and rural areas that are similar to the rest of Sierra Leone.[15] Hence, the sample should be comparable with the rest of the population.

There are several limitations in this study. First of all, we could not measure cholesterol in the total population due to lack of resources. However, online supplementary appendix table 2 shows that there were few differences between the populations with measured cholesterol versus those without cholesterol measurements. The data collection was also limited to within 40 km of Bo City due to accessibility from Bo and travel times. However, all chiefdoms were represented within this distance and were entered into the randomisation. It is unlikely that those areas further from Bo, as an urban centre, would be different from those not selected, as areas more than 40 km from Bo were close to other conurbations in neighbouring districts. We did not control for clustering at household level as few houses supplied more than one participant.

In this study, we have shown that the prevalence of CVDRFs in one of the poorest populations in the world is remarkably high, and the access to care is low. This should have major implications for health policy and planning in Sierra Leone in the years to come. Early deaths and disability due to cardiovascular disease can disrupt the little economic development the country has experienced in recent years and should be given more attention. There is an urgent need to plan where appropriate interventions can be implemented in the most efficient way to make the most of the country's limited healthcare resources, in order to prevent CVDRFs and its consequences.

## CONCLUSIONS

This study shows that about 75% of the population in Bo, Sierra Leone, has at least one cardiovascular risk factor and access to care is very low. In particular, men living in rural areas have a high cardiovascular risk profile and do not access care. The results from this study can inform national plans for cardiovascular disease prevention and management.

## Author affiliations
[1]Institute of Applied Health Research, University of Birmingham, Birmingham, UK
[2]Warwick Medical School, University of Warwick, Coventry, UK
[3]College of Medicine and Allied Health Sciences, University of Sierra Leone, Freetown, Western Area, Sierra Leone
[4]Mercy Hospital Research Laboratory, Bo, Sierra Leone
[5]School of Community Health Sciences, Njala University, Bo Campus, Bo, Sierra Leone
[6]RTI International, Seattle, Washington, USA
[7]Centre for Implementation Science, Health Services and Population Research Department, Institute of Psychiatry, Psychology and Neuroscience, King's College London, London, UK
[8]Department of Biostatistics and Health Informatics, Institute of Psychiatry, Psychology and Neuroscience, King's College London, London, UK
[9]AGE Research Group, NIHR Newcastle Biomedical Research Centre, Newcastle University, Newcastle upon Tyne, UK
[10]Newcastle Upon Tyne Hospitals NHS Foundation Trust, Newcastle Upon Tyne, UK
[11]Centre for Global Surgery, Department for Global Health, Stellenbosch University, Stellenbosch, South Africa

**Acknowledgements** MW acknowledges support from the NIHR Newcastle Biomedical Research Centre. IB is supported by the NIRH Biomedical Research Centre at South London and Maudsley NHS foundation and by the NIHR Collaboration for Leadership in Applied Health Research and Care South London at King's College Hospital NHS Foundation Trust, King's College London. We thank the data collectors (DC) and field manager (FM) who worked on this study for their tireless commitment. These include Ramatu Senesie, DC; Allieu Abu Sheriff, DC; Albert Sidikie Sama, FM; Abdulai Kamara, DC; Umu Binta Bah, DC; Michael Dawson, DC; Christiana Pratt, DC; Michael E. Garrick, DC; Peter Tamba Morsay, DC; Francess Koker, DC; Ismael Vandi, DC; Samuel Kamanda, DC; Wilfred A. U. Jimmy, DC-Team Supervisor; Yvonne Vincentj, DC; Abu Bakarr Mansaray, DC; Mariama Jalloh, DC-Team Supervisor. In addition, we also want to thank and acknowledge the interns (Kadijatu Assiatu Kargbo; Amara Vandi Fomba; Rita Kallon; Veronica Manty Marrah; Carpenter Emmanuel; Bangura A. Ronald; Kpallu Kpakila Sahr; Habibatu Adama Konuwa who supported our research team other research activities).

**Contributors** JD, MW, RN and IB conceived and designed the overall study. JD, TB, HW, RA and JL coordinated baseline data collection and preparation. JD, MW, RN and IB contributed to the design of the household survey. MLO conducted the analysis, and wrote and revised the manuscript. JD supervised the analysis, write up and development of the manuscript. All authors substantively reviewed manuscripts, inputted into revisions and approved the final manuscript.

**Funding** Support for the study was given by the Wellcome Trust, grant number 209921/Z/17/Z.

**Map disclaimer** The depiction of boundaries on this map does not imply the expression of any opinion whatsoever on the part of BMJ (or any member of its group) concerning the legal status of any country, territory, jurisdiction or area or of its authorities. This map is provided without any warranty of any kind, either express or implied.

**Competing interests** None declared.

**Patient consent for publication** Not required.

**Ethics approval** Ethical approval was sought and given from the Sierra Leone Ethical and Scientific Review Committee and the BDM Research Ethics sub-committee at King's College London (HR-17/18–7298).

**Provenance and peer review** Not commissioned; externally peer reviewed.

**Data availability statement** Data may be obtained from a third party and are not publicly available. Data are not publicly available as consent was not given by participants for this to take place.

**ORCID iDs**
Maria Lisa Odland http://orcid.org/0000-0003-4340-7145
Justine Davies http://orcid.org/0000-0001-6834-1838

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
