## [Reviewer comments · BMJ Open]

ARTICLE DETAILS

TITLE (PROVISIONAL)	Prevalence and access to care for cardiovascular risk factors in older people in Sierra Leone: A cross-sectional survey
AUTHORS	Odland, Maria Lisa; Bockarie, Tahir; Wurie, Haja; Ansumana, Rashid; Lamin, Joseph; Nugent, Rachel; Bakolis, Ioannis; Witham, Miles; Davies, Justine

VERSION 1 – REVIEW

REVIEWER	Palash Chandra Banik Bangladesh University of Health Sciences. Bangladesh
REVIEW RETURNED	31-Mar-2020

GENERAL COMMENTS	Abstract, title and references:  [ ] Is the aim clear? Yes [ ] Is it clear what the study found and how they did it? Yes [ ] Is the title informative and relevant? No Explanation: the age of the study participant was 40 years and more but, in the title, they told 'Older' people how they operationalized and what is the reference. It could be the 'adults' instead of 'older'. Are the references:  [ ] Relevant? No Explanation: Some are relevant but some are used unnecessarily. Some of the references were used appropriately like reference no 1 and 2 basically the part of one reference, some of the references used are like this.  [ ] Recent? Yes [ ] Referenced correctly? No Explanation: see above explanation  [ ] Are appropriate key studies included? Some are missing. Introduction/ background  [ ] Is it clear what is already known about this topic? Yes, it is clear. [ ] Is the research question clearly outlined? Yes [ ] Is the research question justified given what is already known about the topic? Yes Attention: the sentence "The reduction in deaths from infections including HIV, together with lifestyle transitions towards a high-calorie, low-activity, urban lifestyle, have already led to a high and rising prevalence of NCDs in lower and middle-income countries (LMIC)." used not clear Methods  [ ] Is the process of subject selection clear? Yes, but have some queries, please see the comments in the pdf manuscript file [ ] Are the variables defined and measured appropriately? No
--

	Explanation: CVD risk estimation was not discussed in the methods section, how they did it in which reference they used. WHO STEPwise approach was not appropriately discussed and reference was missing. The authors only examine the immediate (HTN, DM, dyslipidemia, Obesity) risk factors except for tobacco. But the salt, unhealthy diet and physical inactivity are the very important one which they did not see here. Then in the title and in the objective, they can say the metabolic risk factors. [ ] Are the study methods valid and reliable? No Explanation: how they reach the study subject is not clear enough maybe they can put a map for better understanding in the appendix section. Access to care was not focused on. In the case of sample size, it is very confusing how they determined and why they considered 20% oversample. So many incomplete responses they considered they did not even miss the age data of the eight participants which are a very important nonmodifiable CVD risk factors [ ] Is there enough detail in order to replicate the study? Not at all in this situation. Results [ ] Is the data presented in an appropriate way? Yes [ ] Tables and figures relevant and clearly presented? Yes [ ] Appropriate units, rounding, and the number of decimals? Yes [ ] Titles, columns, and rows labeled correctly and clearly? No Explanation: in case of age they do not put the year or month. In the case of obesity, what was the procedure of measurement BMI or WC? And what was the unit? They put percentage sign unnecessarily with all value. [ ] Categories grouped appropriately? Yes [ ] Does the text in the results add to the data or is it repetitive? No [ ] Are you clear about what is a statistically significant result? Yes [ ] Are you clear about what is a practically meaningful result? Yes Discussion and Conclusions [ ] Are the results discussed from multiple angles and placed into context without being overinterpreted? No, appropriately discussed. [ ] Do the conclusions answer the aims of the study? Yes [ ] Are the conclusions supported by references or results? Yes [ ] Are the limitations of the study fatal or are they opportunities to inform future research? Yes Overall [ ] Was the study design appropriate to answer the aim? Not sufficient [ ] What did this study add to what was already known on this topic? Yes [ ] What were the major flaws of this article? Methods and reference section need substantial improvement [ ] Is the article consistent within itself? Yes Overall statement: Important work but not acceptable in this form for the publication.
--	---

REVIEWER	Alexandra Nowbar Imperial College London, UK
REVIEW RETURNED	21-Apr-2020

GENERAL COMMENTS

Thank you for asking me to review this paper. The study has several merits – it was prospective, there was an appropriate pragmatic sampling technique and the cascade of care concept is useful. There is insufficient information about the regression methodology. For example, why was forced entry of variables chosen? How were the variables chosen? The figures could be improved.

In abstract methods, please amend the grammar “We describe prevalence of CVDRFs diabetes, hypertension...” Should it be “of the following risk factors...”?

In abstract methods, this is unclear, please re-phrase: “Prevalence of CVDRFs and progress through the cascade for hypertension were associated with demographic and socio-economic variables using multivariable regression.” Multivariable regression probably doesn’t need to be mentioned in the abstract. And this seems unnecessary “Reasons for not accessing care were explored”.

In abstract results, please remove “nearly 50%”, please simply report the data. The corresponding CI does not contain the population estimate, please correct this.

In abstract results it is a bit odd to say “regression models showed”, and “people in urban areas were more likely” than who?

In abstract conclusions, please remove “this is the first study to show that” because it’s not really relevant. Typically the conclusion would contain the take home message from your study’s results. Please re-phrase “Health system strengthening with a focus on CVDRFs is urgently needed” as it is a little vague.

Strengths: Please remove being one of the first studies or the first as this isn’t a study strength. You can say elsewhere if you’re the first, perhaps in the discussion. Study strengths would be the way you designed it and/or analysed the data. “Done in a rigorous way to avoid potential biases” is vague, please specify. The geographical limitation could be much more clearly expressed.

Introduction: The first sentence could be clearer and more direct e.g. Cardiovascular disease is the top cause of mortality in the world (disclaimer: <https://www.ncbi.nlm.nih.gov/pubmed/31163980>)

Introduction page 6, line 60: please remove “and overweight”.

Introduction page 7, line 5: sentence too long, please re-phrase into shorter sentences.

Introduction page 7, line 8: Perhaps the authors could elaborate on the “small outdated studies” and could just state the years the data is from, the number of people in the studies and specify the limitations of these studies’ methods.

Introduction page 7, line 10: The first half of this sentence just repeats the points made above. “In sum, there is little rigorous information on the burden of CVDRF in Sierra Leone and no information on whether and how sufferers are accessing care.”

Introduction page 7, line 27: repetition of “access to care”?

	Methods page 7, line 53: "Up to 17.4%" is slightly odd phrasing for the proportion of population over 40. Methods, page 10, line 49: Please make the education categories clearer – "Educational level was defined as no completed education or any education." Methods, page 11: Perhaps the authors could make the cascade of care definitions clearer, especially points 1 and 2. Patient and public involvement Page 12: "Information was fed back to patients if they had abnormal measurements and they were referred to a local health care facility." This statement should be moved to another part of the methods as it is not PPI. Results, page 13: "The final sample included 2071 individuals." Of a possible how many? How many declined? Results, page 13: Please could the weighting be discussed in the methods. Results, page 13: Please amend the CI to include the estimate of hypertension prevalence "49.6% (95% CI 44.1-44.7)" Results, page 14: It may not be necessary for everyone with these cardiovascular conditions to seek healthcare as often as every 3 months so the access to care analysis is overly simplistic. Discussion, page 16 line 5: Please avoid general statements like "rigorously conducted" and be specific about advantages e.g. this was a prospective study. Discussion, page 16 line 9: Regarding this being a study of access to care, this is one of the weaker elements of the study in terms of design so I would de-emphasise this in the discussion. Discussion, page 16 line 13: Please change "about 75%" to the precise number. Discussion: "Less than 20% of the population with hypertension, diabetes and dyslipidaemia accessed health care in the last three months." Yes that sounds low compare to UK standards but isn't necessarily inappropriate. There is no benchmark. I think very limited conclusions can be drawn from the access to care aspect of this study. Discussion page 19, line 32: Please state why cholesterol was not measured in all participants. Conclusions page 20: The first 2 sentences of the conclusions section are not conclusions of this study. Please move to a more appropriate location in the manuscript. The last sentence of the conclusion could be improved - "This study fills a gap in knowledge that is needed to inform" are superfluous words. Figures: Graph titles should be removed because the information should go in the legend. Figure 1: The x-axis labels are too small to read. What is the value in dividing by age group when the pattern does not seem to differ significantly according to age?
--	--

	Figures 2 and 3: As referenced in my comments on the methods, “prevalence” and “screened” in the cascade model are not clear. In my view the black arrows and %change between the vertical bars should be removed as they do not add anything. The figure legend should remind the reader of the definition of the 5 steps of the cascade of care. The figure should be comprehensible to someone who has not read the rest of the paper.
--	--

VERSION 1 – AUTHOR RESPONSE

Reviewer 1:

Abstract, title and references

The age of the study participant was 40 years and more but, in the title, they told ‘Older’ people how they operationalized and what is the reference. It could be the ‘adults’ instead of ‘older’.

Response:

Thanks for this comment. However in Sierra Leone the average life expectancy is 53.9 years and people over 40 are considered old in this context.

Reviewer 1:

References. Some are relevant but some are used unnecessarily. Some of the references were used appropriately like reference no 1 and 2 basically the part of one reference, some of the references used are like this.

Response:

We have updated the references. We hope that this is satisfactory.

Reviewer 1:

Introduction

“The reduction in deaths from infections including HIV, together with lifestyle transitions towards a high-calorie, low-activity, urban lifestyle, have already led to a high and rising prevalence of NCDs in lower and middle-income countries (LMIC).” used not clear

Response:

Thanks for this comment. We have tried to make this sentence clearer please see page 5 line 99 to 102. “The reduction in deaths from infections including HIV has led to an aging population which has together with lifestyle transitions towards a high-calorie, low-activity and urban lifestyle, led to a high and rising prevalence of NCDs in lower and middle income countries (LMIC).”

Reviewer 1:

Methods

CVD risk estimation was not discussed in the methods section, how they did it in which reference they used.

Response:

We are unsure what the reviewer is referring to as we did not estimate the future CVD risk.

Reviewer 1:

WHO STEPwise approach was not appropriately discussed and reference was missing.

Response:

Thank you – we have inserted the reference where the WHO STEPS Survey is mentioned in the

methods section see page 8 line 179-183.

Reviewer 1:

The authors only examine the immediate (HTN, DM, dyslipidemia, Obesity) risk factors except for tobacco. But the salt, unhealthy diet and physical inactivity are the very important one which they did not see here. Then in the title and in the objective, they can say the metabolic risk factors.

Response:

We are focussing on physiological risk factors in this paper. We acknowledge that tobacco is more of a lifestyle factor but it has been included in this paper because it is perceived to have a more direct effect than other lifestyle factors such as salt and diet. The effects of diet and physical activity are explored in a different paper which is under progress.

Reviewer 1:

How they reach the study subject is not clear enough maybe they can put a map for better understanding in the appendix section.

Response:

We have now included a map of Bo where the study was conducted in the appendix.

Reviewer 1:

Access to care was not focused on. .

Response:

We have a now made it clear that access to care was assessed in two ways 1) using the care cascade and 2) self-reported methodology. Please see highlighted text Page 10 Line 218-240.

Reviewer 1:

In the case of sample size, it is very confusing how they determined and why they considered 20% oversample.

Response:

We determined sample size based on previous estimates of diabetes which is the least common risk factor. We considered a 20% oversample because we were worried about lack of willingness to participate after Ebola, and we were uncertain on how the quality of some of the data would be due to the environment of how the data was collected.

Reviewer 1:

So many incomplete responses they considered they did not even miss the age data of the eight participants which are a very important nonmodifiable CVD risk factors.

Response:

Although we estimated we would need 20% oversample the willingness and completeness of the data was good. We only had to remove one participant due to missing data, and regarding age we are only missing data for eight participants. Hence we do not find the there is a lot of missingness in our dataset. We don't know how to further respond to this comment.

Reviewer 1:

There is not enough detail to replicate the study at the moment.

Response:

We have added more information to the methods as requested by the reviewer and hope that there is

now enough detail to replicate the study. We also adjusted the text on selection of chiefdoms were we conducted our study to make it clearer how they were chosen.

Please see page 7, Line 151-155.

“The 15 rural chiefdoms that comprise Bo District were listed in alphabetical order and 7 chiefdoms with separate geographic locations were chosen for the study using random number generator. Settlement groups or villages within these chiefdoms were identified and two were randomly chosen for study. Seven urban communities were randomly selected from 24 urban communities using similar methods of selection.”

Reviewer 1:

Results

In case of age they do not put the year or month.

Response:

Age was defined as years on the last birthday that they passed. Age in years (median with interquartile range) has been added to Table 1. Please see page 22.

Reviewer 1:

In the case of obesity, what was the procedure of measurement BMI or WC? And what was the unit?

Response:

We haven't measured weight circumference. Overweight is a category which is described in the methods page 9 line 198-200 and does not require units. “Body mass index (BMI) was defined as weight (measured in kilograms (kg)) divided by height (measured in meters squared) and classified as normal weight (<25kg/m²) or overweight/obese (BMI ≥ 25kg/m²).” We would normally leave this described in the methods, but if you prefer we can put the category in parenthesis in the table.

Reviewer 1:

They put percentage sign unnecessarily with all value.

Response:

All the unnecessary percentages have been removed see highlighted text page 22 table 1.

Reviewer 1:

Overall impression

Methods and references section need substantial improvement

Response:

The methods and references have been updated according to the reviewer's request. We hope that the changes are sufficient.

Reviewer 2:

Abstract

In abstract methods, please amend the grammar “We describe prevalence of CVDRFs diabetes, hypertension...” Should it be “of the following risk factors...”?

Response:

Thank you for your comment. We have adjusted the text accordingly please see page 3 line 61-63. “We describe prevalence of the following risk factors; diabetes, hypertension, dyslipidaemia, overweight or obesity, smoking and having at least one of these risk factors.”

Reviewer 2:

In abstract methods, this is unclear, please re-phrase: “Prevalence of CVDRFs and progress through the cascade for hypertension were associated with demographic and socio-economic variables using multivariable regression.” Multivariable regression probably doesn’t need to be mentioned in the abstract.

Response:

Thanks for this comment. We feel like we have to keep the name of the statistical test in as a part of the methods, but we have changed the sentence. See page 3, line 66-69. “Multivariable regression was used to test associations between prevalence of CVDRFs and progress through the cascade for hypertension with demographic and socioeconomic variables”

Reviewer 2:

And this seems unnecessary “Reasons for not accessing care were explored”.

Response:

We feel like this is an important part of the methodology, but we have changed the sentence to “In those with recognised disease who did not seek care reasons for not accessing care were recorded.” Please see page 3 line 68-69.

Responding to the reviewers comments has increased the word count in the abstract. This has been truncated in the online submission, but we are happy to edit this down if needed.

Reviewer 2:

In abstract results, please remove “nearly 50%”, please simply report the data.

Response:

Thank you we have adjusted the text and simply reporting the data.

Reviewer 2:

The corresponding CI does not contain the population estimate, please correct this.

Response:

The corresponding CI has been corrected. Please page 3 Line 70-73. “Of 2071 people, 49.6% (95% CI 49.3-50.0)”

Reviewer 2:

In abstract results it is a bit odd to say “regression models showed”, and “people in urban areas were more likely” than who? We have removed regression models.

Response:

We have adjusted the text please see page 3 line 73-74. “People in urban areas were more likely to have diabetes and be overweight than those living in rural areas.”

Reviewer 2:

In abstract conclusions, please remove “this is the first study to show that” because it’s not really relevant.

Typically the conclusion would contain the take home message from your study’s results. Please re-phrase “Health system strengthening with a focus on CVDRFs is urgently needed” as it is a little vague.

Response:

Thanks for your comment. We have removed “this is the first study to show that”.

We have adjusted the text please see page 4 line 79-81 “In Sierra Leone CVDRFs are prevalent and access to care is low. Health system strengthening with a focus on increased access to quality care for CVDRFs is urgently needed.”

Reviewer 2:

Strengths: Please remove being one of the first studies or the first as this isn't a study strength. You can say elsewhere if you're the first, perhaps in the discussion. Study strengths would be the way you designed it and/or analysed the data. “Done in a rigorous way to avoid potential biases” is vague, please specify. The geographical limitation could be much more clearly expressed.

Response:

We had adjusted the strengths and limitations section to only reflect the methodology of the study. Please see page 2 line 33-41 and response to editors.

Reviewer 2:

Introduction: The first sentence could be clearer and more direct e.g. Cardiovascular disease is the top cause of mortality in the world (disclaimer: <https://www.ncbi.nlm.nih.gov/pubmed/31163980>)

Response:

Thank you for this comment, but we would prefer to stick with what we got if possible.

Reviewer 2:

Introduction page 6, line 62: please remove “and overweight”.

Response:

Done as requested please see page 5 Line 118. We have corrected this to keep overweight as this incorporates both overweight and obesity.

Reviewer 2:

Introduction page 7, line 5: sentence too long, please re-phrase into shorter sentences.

Response:

We have adjusted the sentence please see page 5 line 119-122. “Unfortunately, although estimates of CVDRF prevalence from modelling studies exist, very little systematic, direct measurement of the burden of CVDRF in the country has occurred; although small outdated studies have suggested a high burden of CVDRF.”

Reviewer 2:

Introduction page 7, line 8: Perhaps the authors could elaborate on the “small outdated studies” and could just state the years the data is from, the number of people in the studies and specify the limitations of these studies' methods.

Response:

Thanks for this. We have elaborated on the small outdated studies as requested. Please see page 6 line 122-123; “These other studies are either more than 10 years old or have fewer than 700 participants.”

Reviewer 2:

Introduction page 7, line 10: The first half of this sentence just repeats the points made above. “In sum, there is little rigorous information on the burden of CVDRF in Sierra Leone and no information on whether and how sufferers are accessing care.”

Response:

Thanks for this comment. We have removed the first part of the following sentence from the introduction. Please see page 6 line 123-124. "Additionally there is no information on whether and how sufferers are accessing care."

Reviewer 2:

Introduction page 7, line 27: repetition of "access to care"?

Response:

Please see page 6 line 127-130. "In order to provide evidence to assist health policy planning, this study aimed to describe the prevalence of CVDRF in people over 40 years old in Sierra Leone, access to care for those risk factors, and sociodemographic characteristics associated with CVDRF and access to care." Access to care is not repeated. The last part of the sentence is saying we want to look at associations between sociodemographic characteristics and access to care.

Reviewer 2:

Methods page 7, line 53: "Up to 17.4%" is slightly odd phrasing for the proportion of population over 40.

Response:

We have adjusted the text please see page 6 line 140. "17.4% of the population are over 40 years of age".

Reviewer 2:

Methods, page 10, line 49: Please make the education categories clearer – "Educational level was defined as no completed education or any education."

Response:

We have adjusted the text please see page 9 line 210-211; "Educational level was defined as having completed "any level of education" (primary, secondary or University) or "no completed education".

Reviewer 2:

Methods, page 11: Perhaps the authors could make the cascade of care definitions clearer, especially points 1 and 2.

Response:

We have tried to make the cascade of care clearer please see page 10 and line 218-240.

Reviewer 2:

Patient and public involvement Page 12: "Information was fed back to patients if they had abnormal measurements and they were referred to a local health care facility." This statement should be moved to another part of the methods as it is not PPI.

Response:

This section has now been moved to the ethics section. Thanks for this comment.

Reviewer 2:

Results, page 13: "The final sample included 2071 individuals." Of a possible how many? How many declined?

Response:

Our strategy was search and replace and we continued until we had achieved our target sample size.

Very few participants declined to take part and the response rate was close to 100%.

Reviewer 2:

Results, page 13: Please could the weighting be discussed in the methods.

Response:

There is sentence about the weighting in the statistical analysis part of the methods. Please see page 11 line 257-258. "Probability weights for age and sex in Bo-South were calculated based upon the 2015 Population and Household Census."

Reviewer 2:

Please amend the CI to include the estimate of hypertension prevalence "49.6% (95% CI 44.1-44.7)"

Response:

Thanks for noticing this error. The text has been adjusted please see Page 12 Line 277-277; "The prevalence of hypertension was 49.6% (95% CI 49.3-50.0)"

Reviewer 2:

Results, page 14: It may not be necessary for everyone with these cardiovascular conditions to seek healthcare as often as every 3 months so the access to care analysis is overly simplistic.

Response:

As prescriptions are usually only monthly in most low income countries patients need to access care at least once a month.

Reviewer 2:

Discussion, page 16 line 5: Please avoid general statements like "rigorously conducted" and be specific about advantages e.g. this was a prospective study.

Response:

We have adjusted the text to remove rigorously conducted. Please see Page 15 Line 338-340. "This paper reports one of the first studies conducted to provide estimates of the prevalence of all CVDRFs in Sierra Leone; it is the first that we are aware of to publish on access to care for CVDRFs".

Reviewer 2:

Discussion, page 16 line 9: Regarding this being a study of access to care, this is one of the weaker elements of the study in terms of design so I would de-emphasise this in the discussion.

Response:

Access to care is one of the main elements of this paper and is reported in two ways. The cascade of care is especially a recognised method. We tried to make this clearer by adding a separate section on access to care in the methods. Please see Page 10 Line 218-240.

Reviewer 2:

Discussion, page 16 line 13: Please change "about 75%" to the precise number.

Response:

Done as requested. Please see page 15 line 340-341. "Our data suggest that the prevalence of CVDRFs in Sierra Leone is high with 75% of the population over 40 having at least one CVDRF."

Reviewer 2:

Discussion: "Less than 20% of the population with hypertension, diabetes and dyslipidaemia

accessed health care in the last three months.” Yes that sounds low compare to UK standards but isn’t necessarily inappropriate. There is no benchmark. I think very limited conclusions can be drawn from the access to care aspect of this study.

Response:

Actually we have used this method before in larger studies using accepted strong methods and we would disagree that less than 20% of the population is ok. This is a similar conclusion to what other people have drawn. Please see references: <https://pubmed.ncbi.nlm.nih.gov/31327566/>;
<https://pubmed.ncbi.nlm.nih.gov/30822339/>

Reviewer 2:

Discussion page 19, line 32: Please state why cholesterol was not measured in all participants.

Response:

Measurement of cholesterol was not done in all participants due to lack of resources. This has been added to text please see page 18 line 423-424; “First of all, we could not measure cholesterol in the total population due to lack of resources.”

Reviewer 2:

Conclusions

The first 2 sentences of the conclusions section are not conclusions of this study. Please move to a more appropriate location in the manuscript. The last sentence of the conclusion could be improved - “This study fills a gap in knowledge that is needed to inform” are superfluous words.

Response:

We have removed the two first sentences of the conclusions and adjusted the last sentence as requested. Please see page 19 line 444-446. “The results from this study can inform national plans for cardiovascular disease prevention and management”

Reviewer 2:

Figures: Graph titles should be removed because the information should go in the legend.

Response:

We have made the suggested changes. Please see Figures attached.

Reviewer 2:

Figure 1: The x-axis labels are too small to read. What is the value in dividing by age group when the pattern does not seem to differ significantly according to age?

Response:

Age is an important risk factor for cardiovascular disease and all the risk factors differ by age group. Therefore we would like to keep the age groups in the figure. However we can adjust the figure if this is not possible.

Reviewer 2:

Figures 2 and 3: As referenced in my comments on the methods, “prevalence” and “screened” in the cascade model are not clear. In my view the black arrows and %change between the vertical bars should be removed as they do not add anything.

The figure legend should remind the reader of the definition of the 5 steps of the cascade of care. The figure should be comprehensible to someone who has not read the rest of the paper.

Response:

This is the standard way of doing a cascade of care figure and showing the loss to care at each step in the cascade.

We have added information to the figure legend to try and make the five steps of the cascade of care clearer.

VERSION 2 – REVIEW

REVIEWER	Palash Chandra Banik Bangladesh University of Health Sciences, Dhaka, Bangladesh
REVIEW RETURNED	11-Jul-2020

GENERAL COMMENTS	I have satisfied with the revised version and these data are very important for the Sierra Leone population as they have very limited data. For any kind of intervention, data is very important. However, I will suggest removing the % sign from all the values in the appendix table.
--

REVIEWER	Dr Alexandra Nowbar Imperial College London, UK
REVIEW RETURNED	15-Jul-2020

GENERAL COMMENTS	Thank you for this revised manuscript. The majority of my concerns appear to have been addressed.
---